# Stress Impairs Skin Barrier Function and Induces α2-3 Linked *N*-Acetylneuraminic Acid and Core 1 *O*-Glycans on Skin Mucins in Atlantic Salmon, *Salmo salar*

**DOI:** 10.3390/ijms22031488

**Published:** 2021-02-02

**Authors:** John Benktander, Henrik Sundh, Kristina Sundell, Abarna V. M. Murugan, Vignesh Venkatakrishnan, János Tamás Padra, Jelena Kolarevic, Bendik Fyhn Terjesen, Marnix Gorissen, Sara K. Lindén

**Affiliations:** 1Department of Medical Biochemistry and Cell Biology, Institute of Biomedicine, Sahlgrenska Academy, University of Gothenburg, Box 440, Medicinaregatan 9C, 405 30 Gothenburg, Sweden; john.benktander@gu.se (J.B.); abarna258@gmail.com (A.V.M.M.); vignesh.venkatakrishnan@gu.se (V.V.); janos.tamas.padra@gu.se (J.T.P.); 2Department of Biological and Environmental Sciences, University of Gothenburg, 405 30 Gothenburg, Sweden; henrik.sundh@bioenv.gu.se (H.S.); kristina.sundell@bioenv.gu.se (K.S.); 3Nofima, Sjølsengen 22, 6600 Sunndalsøra, Norway; jelena.kolarevic@nofima.no; 4Cermaq Group AS, 0190 Oslo, Norway; bendik.fyhn.terjesen@cermaq.com; 5Radboud University, Institute for Water and Wetland Research, Department of Animal Ecology & Physiology, Heyendaalseweg 135, 6525 AJ Nijmegen, The Netherlands; M.Gorissen@science.ru.nl

**Keywords:** *O*-glycan, skin, barrier function, stress, Atlantic salmon, mucin, Liquid chromatography–mass spectrometry, stress indicator

## Abstract

The skin barrier consists of mucus, primarily comprising highly glycosylated mucins, and the epithelium. Host mucin glycosylation governs interactions with pathogens and stress is associated with impaired epithelial barrier function. We characterized Atlantic salmon skin barrier function during chronic stress (high density) and mucin *O*-glycosylation changes in response to acute and chronic stress. Fish held at low (LD: 14–30 kg/m^3^) and high densities (HD: 50-80 kg/m^3^) were subjected to acute stress 24 h before sampling at 17 and 21 weeks after start of the experiment. Blood parameters indicated primary and secondary stress responses at both sampling points. At the second sampling, skin barrier function towards molecules was reduced in the HD compared to the LD group (P_app_ mannitol; *p* < 0.01). Liquid chromatography–mass spectrometry revealed 81 *O*-glycan structures from the skin. Fish subjected to both chronic and acute stress had an increased proportion of large *O*-glycan structures. Overall, four of the *O*-glycan changes have potential as indicators of stress, especially for the combined chronic and acute stress. Stress thus impairs skin barrier function and induces glycosylation changes, which have potential to both affect interactions with pathogens and serve as stress indicators.

## 1. Introduction

Atlantic salmon aquaculture has grown tremendously the last decades and the yearly world production has exceeded 2 million tonnes since 2012 [1]. In intensive Atlantic salmon (*Salmo salar*) aquaculture, husbandry-related practices such as netting, handling, transportation, and stocking density may be stressful to the fish, which in turn may compromise fish welfare and health [2,3,4,5,6,7,8]. Upon recognition of a stressor, the central nervous system elicits a series of behavioral and physiological responses in order to restore homeostasis [9]. In this primary response, the hypothalamic-pituitary gland-interrenal (HPI) axis is activated and leads to increased circulating levels of the glucocorticoid cortisol [9]. Depending on the severity and duration of the stressor, plasma cortisol can either remain elevated or return to unstressed levels even though the stressor is still present [7,8,10,11,12,13,14]. Therefore, plasma cortisol levels are suggested to be a good indicator of acute stress but a poorer indicator of more long-term or repeated stress [7,15]. The primary stress response leads to secondary stress responses in target tissues. A conserved response to acute and chronic stress in vertebrates is impaired intestinal barrier function manifested as increased translocation of luminal content (e.g., ions, larger molecules and pathogens) and intestinal inflammation [16]. It is currently not known if acute or chronic stress affects skin barrier function of fish.

The fish skin is in direct contact with the external environment and its integrity is essential for fish welfare and health. The outermost tissue layer is the epidermis consisting mainly of a stratified layer of non-keratinized epithelial cells. Tight junctions between the epithelial cells form a coherent intrinsic barrier to protect the fish from harmful substances in the environment. The epidermis also contains goblet cells that produce and secrete the mucus layer that forms the extrinsic physical barrier [17,18]. The secreted mucus layer is mainly composed of glycoproteins called mucins. They consist of a protein backbone with *O*-glycans attached to serine and threonine residues. Next-generation sequencing of salmon skin has detected sequences that are orthologs to parts of the human MUC2 and MUC5 mucins [19]. Mucins are highly glycosylated and 50–90% of their total weight might be composed of carbohydrates [20]. The mucin glycans can be very diverse, and change in response to environmental factors [21], which has been demonstrated, e.g., during soy-bean-induced enteritis in arctic char (*Salvelinus alpinus*) [22], and in mammals during infections [23,24,25]. These changes in glycosylation, governed by glycosyltransferases and substrate availability, in turn affect the interactions between host and pathogen [24,25,26]. In rats, both chronic and acute stress causes tissue specific changes in glycosyltransferase activity [27].

The *O*-glycosylation of salmon skin mucins has been investigated previously [28,29] and encompass relatively short *O*-glycans (2–6 monosaccharides), with core 1, 2, 3, and 5 containing three types of sialic acids: *N*-acetylneuraminic acid (Neu5Ac), *N*-glycolylneuraminic acid (Neu5Gc) and deaminoneuraminic acid (Kdn). The dominating glycan found on the skin is Sialyl-Tn (Neu5Acα2-6GalNAc).

The fish pathogen *Aeromonas salmonicida* binds to sialic acids on salmon mucins, and terminal *N*-Acetylhexosamine (HexNAc) on intestinal mucins promotes *A. salmonicida* growth [30,31]. *A. hydrophila*, *Vibrio harveyi*, *Moritella viscosa*, and *Yersinia ruckeri*, also bind to mucins from Atlantic salmon and Arctic char, although the structures they adhere to are currently not identified [32]. The glycan structures on mucins can thus greatly affect the interaction with pathogens [21].

Given the importance of skin barrier function for fish welfare and health in aquaculture, it is important to understand how stress associated with intensive production protocols in aquaculture changes the skin barrier function and *O*-glycan repertoire. The main aims of the present study were to characterize Atlantic salmon skin intrinsic barrier function and mucin *O*-glycosylation in response to acute and chronic stress. We hypothesized that both acute and chronic stress impairs the skin barrier function and changes the skin mucin *O*-glycosylation pattern. In order to address this hypothesis, Atlantic salmon were subjected to chronic (high density) or acute (30 min crowding) stress (AS). We then quantified key primary and secondary stress indicators (i.e., plasma cortisol, glucose, ions, HCO_3_^−^, and pH) as well as the impact of stress on skin barrier function and skin mucin *O*-glycosylation.

## 2. Methods

### 2.1. Experimental Design

Norwegian Atlantic salmon were obtained from Nofima Centre for Recirculation in Aquaculture (Sunndalsøra, Norway). Post-smolts (AquaGen strain, Neptun Settefisk) with the start weight of 65.1 ± 0.3 g originating from brackish recirculating aquaculture system (12‰ seawater) were transferred and raised in a flow-through system with seawater (32‰) for 25 weeks (182 days). The post-smolts were raised in octagonal 3.3 m^3^ tanks at an average temperature of 13.1 ± 0.4 °C, photoperiod of 12 h light and 12 h darkness. Fish were fed continuously over 24 h in excess (20% overfeeding) with the commercial feed Skretting Spirit supreme (4.5 mm pellet size).

The chronic stress conditions were achieved in three tanks by stocking the fish at a starting density of 50 kg/m^3^; high density, HD. The three tanks acting as controls had an initial stocking density of 14 kg/m^3^; low density, LD. The stocking densities were allowed to increase up to 30 and 80 kg/m^3^ in the LD and HD group, respectively. The biomass was thereafter maintained at these maximum levels by regular biomass removal throughout the experiment. In addition, to ensure stable chronic stress conditions the specific water flow rates were kept at 0.7–0.8 L min^−1^ kg^−1^ for LD group and at 0.3–0.4 L min^−1^ kg^−1^ for HD group. The LD were chosen based on current the recommendation for Atlantic salmon production which is 15–25 kg m^3^ [33,34], while the HD treatment has previously been determined as stressful to Atlantic salmon [8,35]. Water velocity for both groups was between 0.5–0.6 body length/s. Water quality parameters were measured at the tank level on three occasions during the experiment and results are presented in Table 1. Oxygen saturations in the tanks was monitored using real-time oxygen sensors and was kept automatically above 85% [36]. The average oxygen saturation during the whole experiment was 92.8 ± 0.9% for LD group and 94.0 ± 1.3% for HD group. Carbon dioxide concentrations were between 10.7–12.8 mg/L in the LD group and 14.0–22.1 mg/L in the HD group and reflected the difference in the specific flow rate, which was a part of the experimental treatment. Then, 24 hours before each sampling, subsets of fish from LD and HD groups were subjected to an acute stress (AS) protocol where fish were crowded in buckets at ~300 kg/m^3^ (oxygen > 80%) for 30 min and then transferred to six circular 0.5 m^3^ tanks.

### 2.2. Sampling

On the day of sampling, 17 (S1) and 25 weeks (S2) after start of the experiment, skin mucus and plasma were sampled from LD and HD groups before, or 24 h after the AS protocol (LD-AS and HD-AS). Fish were netted and euthanized with an overdose of MS-222. Samples were taken for analysis of blood physiology, plasma cortisol, skin barrier function and morphology, and skin mucus for mucin isolation and analysis as described in detail below.

### 2.3. Blood Physiology

Whole blood was sampled from the caudal vessels using Vacuette^®^ heparinized vacuum tubes (Greiner Bio-One, Kremsmuster, Austria) and were instantly analyzed using an i-STAT Portable Clinical Analyzer with EC8+ disposable cartridges (Abbott Laboratories, Chicago, IL, USA) for sodium (Na^+^), glucose (Glu), bicarbonate (HCO_3_^−^) and pH. Measured pH was temperature corrected, as previously described [37].

### 2.4. Plasma Cortisol

Plasma cortisol was analyzed using a custom ELISA, as previously described [38], with the only difference that 10 µL undiluted plasma was assigned in duplicate to designated wells. The detection limit was 0.4 ng/mL plasma and samples below detection limit were set to detection limit.

### 2.5. Skin Barrier Function and Morphology

Skin samples were taken from individual fish at the last sampling point, for assessment of electrophysiological and permeability parameters as measure of the barrier function. Fish were carefully netted and euthanized as described above followed by a sharp blow to the head. The fish was placed with the right side up and skin samples 2 × 2 cm were cut from the dorsal skin just below the dorsal fin using scissors, gently stripped from muscle tissue, and placed in salmonid Ringer’s solution (in mM; NaCl 150, KCl 2.5, MgSO_4_ 1, CaCl_2_ 2.5, HEPES 5 and glucose 10) saturated with air on ice. Skin samples were then mounted in modified Ussing chambers [39], filled with Ringer (4 mL per half-chamber). Transepithelial electrical resistance (TER), transepithelial potential difference (TEP) and short-circuit current (SCC) were determined under iso-osmotic conditions according to Sundell et al., 2003 [40] with modifications described by Sundell and Sundh, 2012 [41]. After mounting, the tissue was allowed 60 min acclimation in the chamber after which the experiment started (t = 0). Then the Ringer solution in the dermal chamber was replaced with fresh Ringer only, while the epidermal chamber was replaced with fresh Ringer containing ^14^C-mannitol (42 kBq mL^−1^), a well-defined marker molecule for paracellular transfer in fish [41] and mammals [42]. Electrical parameters were measured every 5 min. A 100 μL sample was withdrawn from the dermal side and replaced with fresh Ringer to avoid changes in hydrostatic pressure at 0, 20, 25, 30, 60, 80, 85, 90 min placed in a scintillation vial filled with 5 mL of scintillation liquid (Ultima GoldTM; PerkinElmer Inc., Waltham, MA, USA). Radioactivity was measured using a β-counter (Wallac1409 DSA liquid scintillation counter; PerkinElmer Inc, Sollentuna, Sweden). The apparent permeability coefficient for mannitol, *P_app_*_,_ was calculated using the following Equation (1):(1)$$P_{app}\;=\;dQ/dT\;\times \;1/AC_{o}$$
where *dQ*/*dT* is the appearance rate of the molecule in the serosal compartment of the Ussing chamber, *A* is the area of intestinal surface exposed in the chamber (0.75 cm^2^), and *C_o_* is the initial concentration (mol mL^−1^) on the mucosal side.

Skin samples for assessment of epidermal thickness were taken from the dorsal skin just below the dorsal fin using scissors and fixed in 4% buffered formalin 24 h and stored in 70% EtOH until further processing. Skin samples were embedded in paraffin wax using standard protocol. The embedded samples were sectioned at 6 µm thickness using a Finesse ME microtome (Labex Instrument, Helsingborg, Sweden). The sections were mounted onto glass slides coated with 3- Aminopropyltriethoxysilane (APES; Sigma–Aldrich, Stockholm, Sweden) and stained with haematoxylin (Harris), eosin, alcian blue. The sections were photographed with a DXM1200 camera (Nikon, Tokyo, Japan) mounted on a Eclipse E1000 microscope (Nikon, Tokyo, Japan) with a 40× objective. The mean epidermal thickness was determined for each fish.

### 2.6. Mucus Sampling and Mucin Isolation

Euthanized fish were placed on the left ventral side, while the right ventral side was scraped with a glass-slide, collected and placed in a 2-mL eppendorf tube with 10 mM sodium phosphate buffer, pH 6.5, with 0.1 mM phenylmethylsulfonyl fluoride (PMSF). The samples were frozen on dry-ice and stored at −80 °C until further treatment. A crude mucin extract was obtained by placing the salmon skin mucins scrapings into twice the sample volume of extraction buffer (6 M guanidine hydrochloride (GuHCl), 5 mM EDTA, 10 mM sodium phosphate buffer, 0.1 mM PMSF, pH 6.5) and homogenizing using a Dounce homogenizer with four strokes with a loose pestle. The solution was transferred to a new tube together with another two sample volumes of extraction buffer used to rinse the homogenizer. The tubes were put on a rocking board for 20 h at 4 °C. The samples were then centrifuged at 3900× *g* for 80 min. Supernatants were collected and saved as crude mucin fractions at 4 °C. The mucins were purified on an in-house prepared one-dimensional sodium dodecyl sulfate-polyacrylamide gel electrophoresis (SDS Ag-PAGE) and electroblotted to a PVDF membrane (ImmobilonPSQ) using a semidry transfer method, as previously described [43].

### 2.7. O-Glycan Analysis by LC-MS

Mucin bands on the PVDF membranes were visualized by Alcian blue (Sigma–Aldrich, St. Louis, MO, USA) staining. Blue bands were excised and subjected to reductive β-elimination. In brief, bands were incubated with 1 M NaBH_4_ and 50 mM NaOH for 16 h at 50 °C. Reactions were quenched with glacial acetic acid, the samples were desalted with AG50WX8 cation-exchange resin (Bio-Rad) and placed in a reversed-phase μ-C18 ZipTip (Millipore). The eluate was dried in a SpeedVac and methanol was added and dried a total of five times to evaporate borate remaining in the sample. Take note that reductive β-elimination likely would remove ester-bonds of *O*-acetylated sialic acids.

After the release and isolation of the reduced *O*-glycans, they were analyzed by liquid chromatography–mass spectrometry. In the liquid chromatography an in-house packed 10 cm × 250 μm column, containing 5 μm porous graphitized carbon (PGC) particles (Thermo Scientific, Waltham, MA, USA), were used. The glycans were eluted with a 0–40% gradient of acetonitrile in 10 mM ammonium bicarbonate over 40 min. with a flow rate of 10 μL/min. The *O*-glycans were detected with an LTQ mass spectrometer (Thermo Scientific) in negative-ion mode. An electrospray voltage of 3.5 kV, capillary voltage of −33.0 V, and capillary temperature of 300 °C were used with compressed air used as a sheath gas. A mass range of *m*/*z* 380–2000 was used and MS/MS was made at a normalized collisional energy of 35%. The isolation width was set to *m*/*z* 2.0 with an activation time of 30 ms and a minimal signal threshold of 300 counts for MS/MS. The data were processed using Xcalibur software (version 2.0.7, Thermo Scientific, Waltham, MA, USA). Glycans were annotated from their MS/MS spectra manually and validated by available structures stored in Unicarb-DB [44] database if possible.

### 2.8. Statistical Analyses

Blood and plasma parameters were analyzed using 2-way ANOVA with fish density and acute stress as fixed factors. In case of significant interaction effects a Sidak-corrected multiple comparisons post hoc test was used for further analyses. Skin barrier function and epidermal thickness was analyzed using independent Student’s *t*-tests. Kruskal–Wallis test followed by Dunn´s post hoc test were used to compare the relative abundances of glycan structures between groups. Differences with *p* ≤ 0.05 were considered significant.

## 3. Results

### 3.1. Effects of Chronic and Acute Stress on Plasma Parameters

Plasma cortisol levels and glucose were assessed as indicators of a primary stress response, while plasma sodium levels was used as indicator for osmoregulatory status. In addition, blood pH and HCO_3_^−^ levels were measured to assess potential impact of acute and chronic stress on pH regulation. At 17 (S1) and 25 (S2) weeks after the start of the experiment four groups of fish were sampled: low density without AS (LD), low density 24 h after acute stress (LD-AS), high density without AS (HD), and high density 24 h after AS (HD-AS). At S1, there were no statistical differences in plasma cortisol levels between LD and HD fish (Figure 1A). However, samples below detection limit were more frequent in the LD group (*n* = 7) compared to the HD group (*n* = 2). After the AS, plasma cortisol levels were elevated in both the LD and HD groups (*p* < 0.05). Plasma glucose levels were increased after AS (*p* < 0.0001) whereas no interaction or effect of chronic stress were observed (*p* > 0.05; Figure 1B). Plasma Na^+^ levels were higher in the HD compared to the LD group (*p* < 0.05) while acute stress resulted in elevated Na^+^ levels in the LD group only (*p* < 0.0001; Figure 1C). At S2, the HD group displayed elevated plasma cortisol levels (*p* < 0.05; Figure 1D), but only the LD group showed an increase in plasma cortisol levels 24 h after AS. Plasma glucose increased after AS (*p* < 0.0001) whereas no interaction or effect of chronic stress were observed (*p* > 0.05; Figure 1E). The AS protocol resulted in elevated blood Na^+^ levels in both LD and HD (*p* < 0.01), while no difference was observed between density groups (Figure 1F).

A significant interaction was observed for blood HCO_3_^−^ at S1 and S2 (*p* < 0.0001; Figure 2A,C). The following pots hoc analyses revealed that at S1, HCO_3_^−^ levels was increased in the HD group as compared to the LD group (*p* < 0.0001). After AS treatment there was a decrease in HCO_3_^−^ levels that was more prominent in the HD group (*p* < 0.0001) compared to the LD group (*p* < 0.05). A similar pattern was observed at S2, with higher HCO_3_^−^ levels in the HD compared to the LD group (*p* < 0.001). Whereas at this time point, AS decreased the HCO_3_^−^ levels similarly in both groups (LD *p* < 0.0001and HD *p* < 0.0001).

A significant interaction was also observed for blood pH at S1 (*p* < 0.001; Figure 2B). The pH was higher in the HD group (*p* < 0.001) and decreased after AS (*p* < 0.001). At S2, the HD group had overall higher pH (*p* < 0.05). After AS, pH decreased in both LD and HD group (*p* < 0.001, Figure 2D).

### 3.2. Effects of Chronic Stress on Skin Barrier Morphology and Function

Chronic stress has previously shown to reduce intestinal barrier function towards ions and uncharged molecules in fish, as well as mammals. We therefore assessed the skin barrier function of Atlantic salmon towards ions (TER) and mannitol (*P_app_*). The diffusion rate for ^14^C-mannitol across the skin was higher in the HD group compared to LD (*p* < 0.01) showing decreased barrier function towards small uncharged molecules (Figure 3A). Similarly, there was a tendency towards a reduction in TER in the HD group, even though not statistically significant (*p* = 0.07; Figure 3B). The epidermis of the HD group was thinner than in the LD group (*p* < 0.001, Figure 3C).

### 3.3. LC-MS of Atlantic Salmon Skin O-Glycans Detected 81 Structures

*O*-glycan structures from mucins were examined by LC-MS/MS from S1 and S2 (LD/HD and 24 h AS), from both low and high-density (LD (*n* = 11 + 6), LD-AS (*n* = 5 + 6), HD (*n* = 5 + 5), HD-AS (*n* = 6 + 6)). For space efficiency reasons, we first present the overall skin glycan repertoire and identify *O*-glycan structures that were altered by stress regardless of size or time in salt water by pooling data from S1 and S2. Thereafter we present glycans that were affected differently by stress depending on sampling time point. Finally, we highlight the structures that may have potential as biomarkers for stress.

A total of 81 structures were identified (Appendix A), of which 25 are described in the Atlantic salmon skin for the first time [28,29]. Only five out of 81 structures were found in all fishes examined while 47 structures were found in at least one fish from each treatment group. The five structures found in all fish comprise on average 79% of the relative abundance of the glycan structures and the other 76 structures are thus of relatively low abundance. The relative abundances of the 25 most abundant structures are shown in Figure 4. In line with earlier studies [28,29], sialyl-Tn (NeuAcα2-6GalNAcol) was the dominating glycan (average 47.6%). The levels of Sialyl-Tn (at *m/z* 513.2) were 20.8% lower (*p* = 0.010) in the HD fish than in the LD fish, with a significant decrease in HD-AS compared to LD fish (Figure 4A). Furthermore, the sialylated core 1 structures at *m/z* 675.3b (Galß1-3[NeuAcα2-6]GalNAcol), 966.3 (NeuAcα2-3Galß1-3[NeuAcα2-6]GalNAcol) and 1024.4a (Fuc-HexNAc-Galß1-3[NeuAcα2-6]GalNAcol) were increased in the stressed fish with significant differences between the LD control and the HD-AS fish group (Figure 4A,B).

### 3.4. Core 1, 2, 3, and 5 Were Detected and Skin Mucins Express More Core 1 Structures When Atlantic Salmon Are Subjected to Both Chronic and Acute Stress

The type of core structures synthesized is a determining factor in what structures the available glycotransferases are able to synthesize. The core structures found are dependent on tissue type and expression can change due to, e.g., infection and inflammation [23]. In total, four *O*-glycan core structures were detected in the skin mucins: core 1, 2, 3, and 5. Of these, core 1 and 5 were most abundant (median 7 and 10% relative abundance, respectively). Although the abundant Sialyl-Tn structures (NeuAcα2-6GalNAcol, NeuGcα2-6GalNAcol and Kdnα2-6GalNAcol: median 76% relative abundance) are strictly not core structures, they were included in this calculation. The relative abundance of core 1 structures was significantly increased in the HD-AS cohort compared to the LD control and LD-AS fish (Figure 5). The single stressed (acute: LD *v/s* LD-AS and chronic: LD *v/s* HD) groups also displayed a modest trend towards an increase in core 1 structures.

### 3.5. Fish Subjected to Both Chronic and Acute Stress Have a Higher Proportion of Large O-Glycan Structures

Increased glycan size can be associated with decreased mucin biosynthesis rate due to longer time spent in the biosynthesis machinery in the Golgi [46]. Furthermore, large glycans allow the terminal structures to be more accessible to microorganisms due to additional reach. The *O*-glycan structures detected by LC-MS were categorized by monosaccharide length for comparison. The majority of glycans found were composed of two monosaccharides and mainly constituted by sialyl-Tn, but structures comprised of up to eight monosaccharides were detected. The largest abundance of long and complex *O*-glycans (more than 4 monosaccharides) was present among the HD-AS group (*p* < 0.05, Figure 6).

### 3.6. Fish Subjected to Both Chronic and Acute Stress Have an Increased Relative Abundance of Structures with Terminal Hexoses and α2-3 Linked NeuAc

The terminal moieties of the *O-*glycans are the main interaction target for pathogens in mucins and are used as a decoy to prevent direct adhesion to the epithelial cells, but can also be used by the pathogens as a nutrient source and to colonize the mucus layer [21,30]. Significantly more Hexoses and α2-3 linked NeuAc were found in the HD-AS cohorts, while a trend towards a higher ratio of NeuGc/NeuAc was observed for the HD fish (Figure 7). The hexoses are most likely mainly composed of galactose, based on similar MS/MS spectra and retention times to previously identified *O*-glycan structures. The level of sulfation on the glycans was overall very low, but increased in response to stress (Figure 7). The largest difference was in NeuAcα2-3, where the median relative abundance in the HD-AS cohort was more than 3 times higher than the median value of the LD cohort (*p* < 0.01, Figure 7).

### 3.7. Some Glycans Changed Differentially Depending on Sampling Time Point

Despite being raised in similar conditions, *O-*glycan differences between the S1 and S2 groups were detected. In the 25 weeks post transfer to seawater (S2), the fucosylated structures were increased in fish subjected to both chronic and acute stress (HD-AS), while this change was not observed at 17 weeks post transfer (S1, Figure 8A). In addition, structures with terminal HexNAcs decreased in S1 when subjected to the combination of acute and chronic stress while in S2 this was not the case (Figure 8B). However, the terminal HexNAc levels tended to be lower after AS regardless of fish density and time point. When pooling the results from all time points and conditions, the fish subjected to the AS protocols showed a significant decrease in HexNAcs compared to their corresponding “control” i.e., the matching LD/HD and/or S1/2 group (Figure 8C).

The difference between mucin *O*-glycans from acutely and chronically stressed fish was subtle. Overall, chronic stress seemed to affect the glycosylation more than AS, though the compound chronic and AS affected glycan expression most. Although the compound chronic and acute stress affected the glycan expression most clearly, this was not always the case, as some glycan structures such as those with HexNAc terminal structures and core 5 (mainly GalNAcα1-3[NeuAcα2-6]GalNAc-, *m/z* 716) tended to decrease by AS and either increase or decrease in chronic stress depending on the post transfer to seawater time point (S1 and S2).

### 3.8. Some O-Glycan Structures Have Potential as Stress Biomarkers in Atlantic Salmon

Galß1-3(NeuAcα2-6)GalNacol and NeuAcα2-3Galß1-3(NeuAcα2-6)GalNAcol both increase in fish subjected to both chronic and AS, i.e., the HD-AS group (Figure 9A,B). Galß1-3(NeuAcα2-6)GalNacol levels appear to be an especially good indicator for distinguishing this double-stressed group from the other groups. A more general approach could be to examine the percentage of the NeuAc containing structures that are NeuAcα2-3 structures (Figure 9C) or the core type 1 structures (Figure 9D). These structures were affected similarly at S1 and S2, suggesting that these changes are stable over time. All cohorts except the LD control group showed high inter-individual variation in the levels of core 1 structures, which could indicate that individual fish in the cohorts might react differently to the stress. No core 1 relative abundance above 11% were detected in the LD control, while relative abundances up to 40% were detected in the other groups (Figure 9D). A similar situation was seen for structures with α2-3 linked NeuAc, where no LD control samples were above 6% relative abundance whereas the other three groups had individual fish with relative abundances up to 35% (Figure 9C).

## 4. Discussion

Here we characterized how Atlantic salmon skin epithelial barrier function changes in response to chronic stress, and changes in mucin *O*-glycosylation in response to acute and chronic stress. Long-term rearing in HD resulted in disturbed osmoregulation and acid–base balance, reduced the thickness of the epidermis concurrent with impaired skin barrier function towards ions and small molecules. In addition, the mucins of the mucus layer were affected by chronic and acute stress. LC-MS revealed 81 structures, expanding the known Atlantic salmon skin mucin *O*-glycome by 22%. Fish subjected to both chronic and AS had an increased proportion of large *O-*glycan structures and some of the *O*-glycosylation changes that occurred in the most severely stressed group (HD-AS) have potential as a non-invasive biomarkers for accumulative stress.

The observed reduction in skin barrier function in the HD group at S2 suggest that this environment was stressful to the fish, even though no significant differences were observed in plasma cortisol or glucose levels. These observations are in agreement with previous findings of secondary stress responses (e.g., reduced intestinal barrier function), in absence of a measurable primary stress response [7,8]. Thus, reduced epithelial barrier function may be a general consequence of chronic stress. Further, reduced skin barrier function will result in increased influx of ions and thus increase in osmoregulatory load. Despite reduced barrier function of the skin, HD fish were able to osmoregulate, as plasma Na^+^ levels were in the same range in LD and HD groups indicating that the HD environment appears to be within their scope of acclimation, but impose an increased allostatic load, and thus a higher energetic cost. In addition, the reduced skin barrier function in the HD group may imply that also harmful substances including antigens, toxins, or pathogens could get easier access to the fish during chronic stress. Collectively, these results show that a HD environment is stressful and imposes an allostatic load to the fish and provide an explanation to increased disease susceptibility and reduced growth in the stressful HD husbandry environments.

Plasma HCO_3_^−^ levels were similar at both sampling time points but were elevated in the HD group, which was corroborated by higher plasma pH. Plausible explanations for these observations may be that the increased HCO_3_^−^ levels in the HD group is a result of increased CO_2_ exposure and later increased metabolism and CO_2_ excretion. At the first sampling time point, the increased HCO_3_^−^ levels is most probably a result of the increased water CO_2_ levels, as positive correlation has previously been observed between plasma HCO_3_^−^ levels and water CO_2_ levels in Atlantic salmon [47]. However, at the last sampling the CO_2_ levels in the HD group were not significantly higher than the LD group. At this time point, the elevated blood HCO_3_^−^ levels may instead be a result of an increased metabolism and CO_2_ excretion due to increased stress. This is supported by Brauner et al. (2000) [48] who showed that blood HCO_3_^−^ levels increased with increasing exercise intensity in seawater acclimated rainbow trout, probably through increased activity of the HCO_3_^−^/Cl^−^ exchanger. Increased activity of the HCO_3_^−^/Cl^−^ exchanger, and thus increased Cl^−^ efflux, would also help compensating for potential osmoregulatory stress induced by a leaky skin barrier. During acute stress, fish are predicted to produce an abundance of lactate creating a decrease in plasma pH, which will be maintained for an extended period of time [49]. Consequently, the clear drop in blood pH, 24 h after subjecting the fish to the AS protocol is probably the result of lactate and H^+^ produced during AS. Interestingly, both LD and HD fish appear to utilize their blood buffering capacity following AS as seen in the subsequent decrease in HCO_3_^−^ levels and the ability of the LD fish at the first sampling point to maintain control plasma pH.

Mucin glycosylation can govern bacterial adhesion, virulence and growth [22,25,26,30,31,50,51,52], and glycosylation changes modulate host-bacterial interactions [23,25]. Consequently, the stress-induced glycosylation changes have the potential to affect the sensitivity towards infection, as well as the normal microbiota. Indeed, the composition of the skin bacterial community was altered 24 h after acute stress in Atlantic salmon [53]. In addition to glycosylation changes, the mucin production rate can also affect the interactions with microorganisms. A mammalian pathogen has been shown to decrease the mucin production rate, thereby creating a more stable niche for itself [54]. Although metabolic labelling experiments are needed to demonstrate such a decreased mucin production rate [55], an increased proportion of large glycan structures has previously been associated with decreased mucin production rate in mice [46]. The increase in large glycan structures were hypothesized to be a consequence of the fact that a slow mucin production will cause the mucins to spend more time in the Golgi, which would allow for more extensive glycosylation [46]. Along these lines, the increased proportion of large glycan structures among fish subjected to both chronic and AS may indicate a decreased mucin production [46], which may contribute to the increased disease susceptibility during stress in Atlantic salmon. An increased size of mucin *O*-glycans has also been found in the rat colon after water avoidance stress [43], suggesting that this is a conserved response to stress.

The largest differences in mucin glycan expression were found between the LD and HD-AS fish with increased core 1 and α2-3 linked NeuAc structures in the stressed cohorts. This suggests that in contrast to plasma parameters of stress, the glycosylation responds to the accumulated stress from both chronic and acute stressors. The most abundant *O*-glycan with differences between the LD and HD-AS group was NeuAcα2-3Galß1-3(NeuAcα2-6)GalNAcol (*m/z* 966.3) which median is almost five times the relative abundance in the HD-AS cohort compared to the LD. Interestingly, the relative abundance of fucosylated structures only increased after stress in the S2 fish while they decreased in the S1 fish. The relative abundance of terminal HexNAc structures was also differed at the two sampling time points. However, the level of terminal HexNAcs was decreased in the AS groups compared to their corresponding controls at both sampling time points. Differences between S1 and S2 could be due to differences in size or the time spent adapting to the marine environment. Recently, it was shown that the epidermis of Atlantic salmon skin increased in thickness concurrent with more mucus producing cells between one to four month post transfer to seawater [56]. However, no observations were made after the 4 month point. The discrepancies between S1 and S2 may also be explained by the longer period of chronic stress for the S2 cohorts, which is supported by the different responses in plasma parameters between S1 and S2.

An observation made when comparing stressed fish to controls was that not all fish showed an increased level of core 1, NeuAcα2-3 and fucose. This could suggest that not all fish react the same way towards the stress which is in line with a behavioral study on salmon that concluded that “proactive” fish display a higher post-stress serotonergic activity in the proposed amygdala homologue, increased neuroplasticity marker brain-derived neurotrophic factor (*bdnf*) and lower levels of cortisol compared to “reactive” Atlantic salmon [57]. This is further supported by a study on rainbow trout (*Oncorhynchus mykiss*), where dominant fish had lower plasma cortisol levels after a short period of confinement stress compared to subordinate fish [58].

The results of increased NeuAcα2-3 levels in the skin mucus of stressed salmon indicate that it could potentially be used as an indicator of accumulative acute and chronic stress for which reliable physiological indicators are valuable. Core 1 content could also be used, but may have technical drawbacks due to not being a terminal moiety like NeuAc. Specific *O*-glycan structures such as Galß1-3(NeuAcα2-6)GalNAcol and NeuAcα2-3Galß1-3(NeuAcα2-6)GalNAcol also seem to be indicators of compound acute and chronic stress in the fish skin mucins, while the percentage of NeuAcα2-3 to total NeuAc also significantly increases in response to acute and chronic stress. Assays using specific glycan structures as markers would be easier to set up as high throughput assays than the proportion of NeuAcα2-3 to total NeuAc, however, the latter is least likely to be affected by genetic factors. A possible way to get around this issue would be to select several NeuAc containing structures as markers.

The related species Arctic charr (*Salvelinus alpinus*) displays increased levels of NeuGc in response to inflammation in the gastrointestinal tract mucins [22]. Here, we found a trend towards increased NeuGc *O*-glycan structures on Atlantic salmon skin mucins from stressed fish, suggesting that an increase in NeuGc *O*-glycan structures may be associated with a broad range of stressors.

In summary, HD as a chronic stressor reduced skin barrier function. LC–MS revealed 81 *O*-glycan structures from the skin, expanding the known Atlantic salmon skin mucin *O*-glycome by 22%. Further, acute and chronic stress induced skin mucin *O*- glycosylation changes in Atlantic salmon, especially in fish subjected to both chronic and AS. Among the glycosylation changes identified, the increased proportion of large glycan structures suggests a decreased mucin production [46], which may contribute to the increased susceptibility to disease of stressed fish. The structural changes also have potential to affect host pathogen interactions [22,25,26,31,32,52]. Finally, the relative abundance of *O*-glycan structures Galß1-3(NeuAcα2-6)GalNacol, NeuAcα2-3Galß1-3(NeuAcα2-6)GalNAcol and core type 1 structures as well as the proportion of sialylation in the form of NeuAcα2-3 has potential as biomarkers for stress in fish, especially for the accumulative chronic and AS, as this was not apparent in any of the other stress markers. These structures warrant further investigation into their specificity as stress indicators.

## Figures and Tables

**Figure 1 ijms-22-01488-f001:**
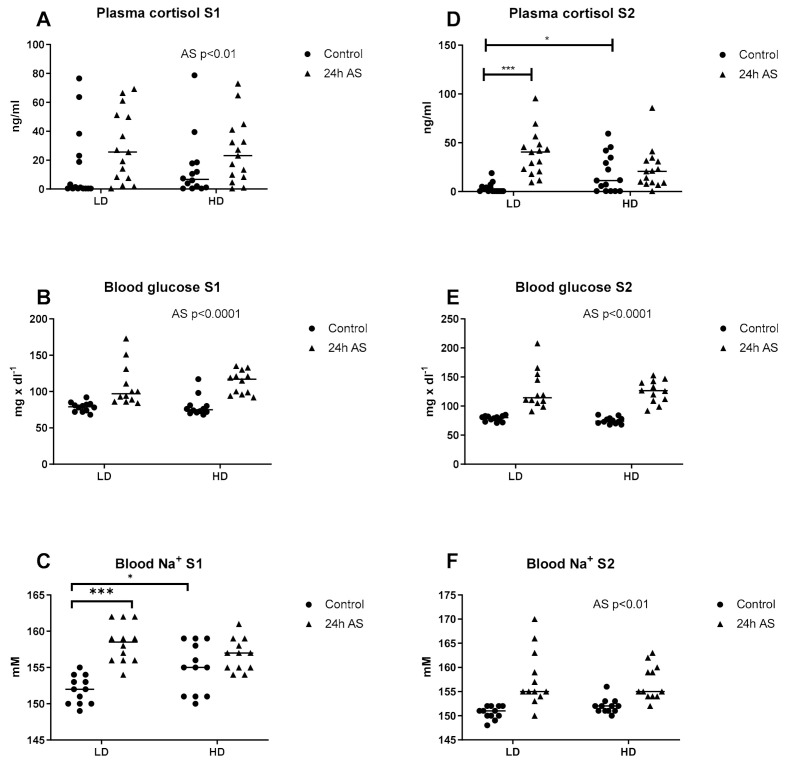
Effect of acute and chronic stress on blood parameters. Atlantic salmon were reared in low density (LD) and high density (HD) conditions with and without being subjected to an acute stress (AS) protocol at two sampling points (S1 and S2) and plasma cortisol (**A**,**D**), blood glucose (**B**,**E**) and blood Na^+^ levels (**C**,**F**) were analyzed. Statistics: 2-way ANOVA where a significant interaction were subjected to Sidak-corrected multiple comparisons. Main effects of density and AS are indicated by text in the figures and significant effects from the post-hoc test are reported as * *p* < 0.05 and *** *p* < 0.001. Data are presented as individual values with the line representing the mean.

**Figure 2 ijms-22-01488-f002:**
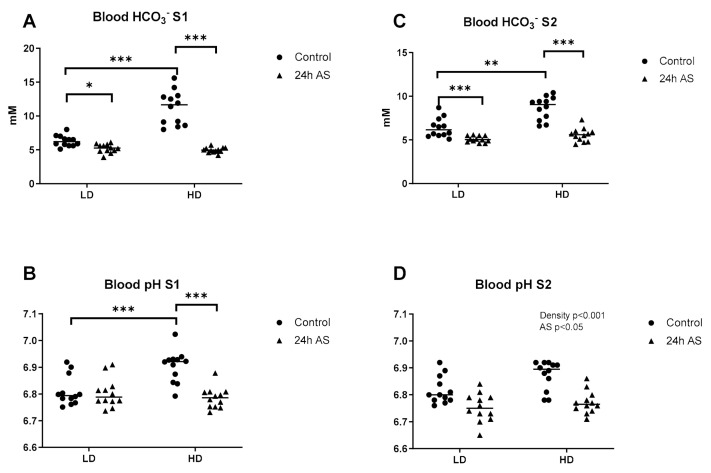
Effects of acute and chronic stress on blood pH and HCO_3_^−^ levels. Atlantic salmon were reared in low density (LD) and high density (HD) conditions with and without being subjected to an acute stress (AS) protocol at two sampling points (S1 and S2) and blood pH (**A**,**C**) and blood HCO_3_^−^ levels (**B**,**D**) were analyzed. Statistics: 2-way ANOVA where a significant interaction were subjected to Sidak-corrected multiple comparisons. Main effects of density and acute stress (AS) are indicated by text in the figures and significant effects from the post-hoc test are reported as * *p* < 0.05, ** *p* < 0.01 and *** *p* < 0.001. Data are presented as individual values with the line representing the mean.

**Figure 3 ijms-22-01488-f003:**
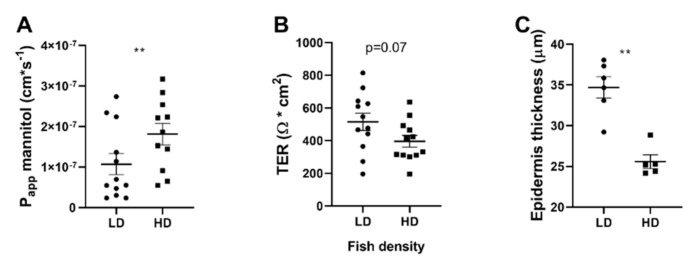
Effects of chronic stress on skin barrier function. Atlantic salmon were reared in low density (LD) and high density (HD) conditions and skin barrier function was assessed as (**A**) Papp for mannitol, (**B**) TER and (**C**) epidermal thickness. Statistic: Student’s *t*-test, ** *p* < 0.01. Data are presented as individual values with the line representing the mean.

**Figure 4 ijms-22-01488-f004:**
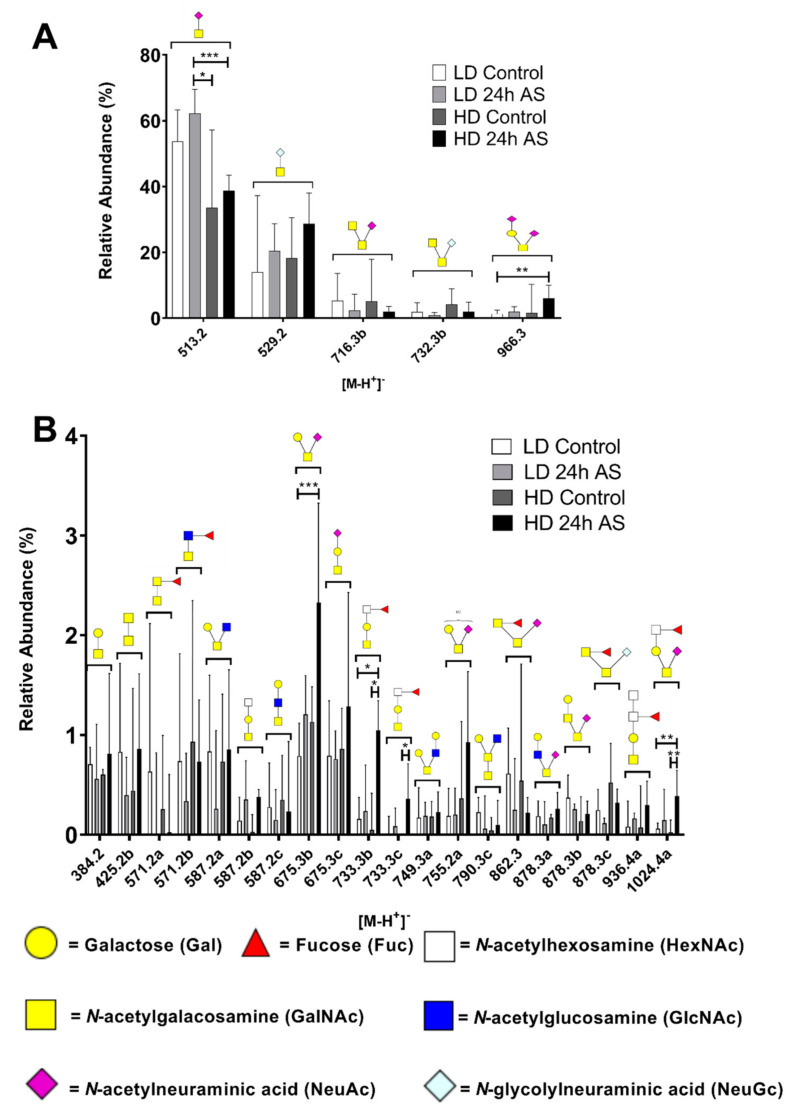
Relative abundance of the 25 most abundant skin mucin *O*-glycans. Atlantic salmon were grown under low density (LD) and high density (HD) conditions with and without being subjected to an acute stress (AS) protocol. Glycans were analyzed by LC-MS. (**A**) High relative abundance *O-*glycans and (**B**) Low relative abundance *O-*glycans. Structures are displayed using Symbol Nomenclature for Glycomics (SNFG) [45]. Available linkages can be found in Table 1. Statistics: * *p* < 0.05, ** *p* < 0.01, *** *p* < 0.001; Kruskal–Wallis test with Dunn’s post hoc test. Data are presented as median and interquartile range.

**Figure 5 ijms-22-01488-f005:**
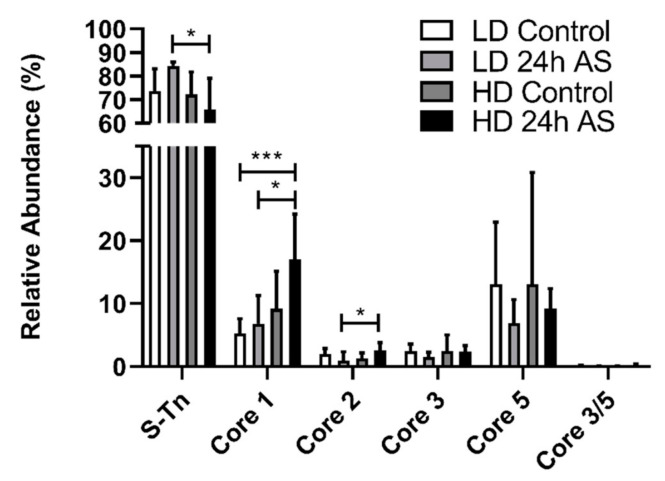
Relative abundance of skin mucin *O*-glycan core structures. Atlantic salmon were grown under low density (LD) and high density (HD) conditions with and without being subjected to an acute stress (AS) protocol and glycans analyzed by LC-MS. Statistics: * *p* < 0.05 and *** *p* < 0.001; Kruskal–Wallis test with Dunn’s post hoc test. Data are presented as median and interquartile range.

**Figure 6 ijms-22-01488-f006:**
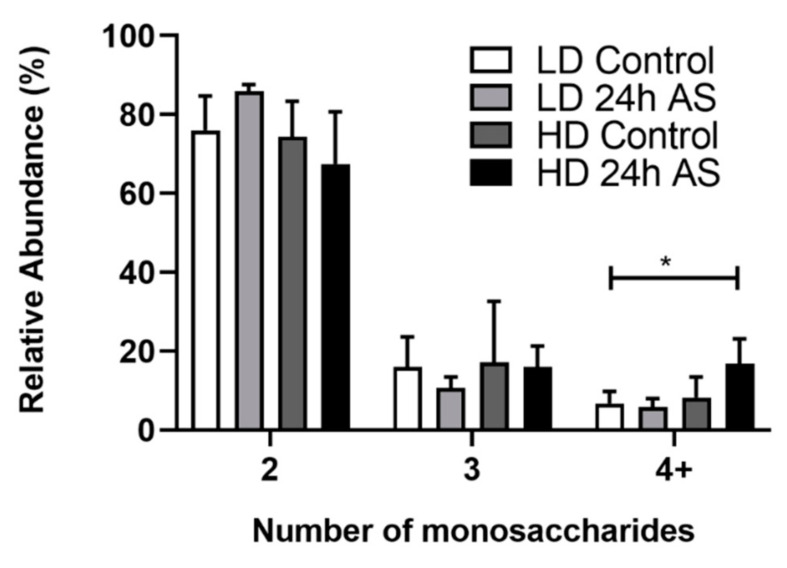
Size distribution of skin mucin *O*-glycans. Atlantic salmon were grown under low density (LD) and high density (HD) conditions with and without being subjected to an acute stress (AS) protocol and glycans analyzed by LC-MS. The size is presented according to the number of monosaccharides in each structure. Statistics: * *p* < 0.05; Kruskal–Wallis test with Dunn’s post hoc test. Data are presented as median and interquartile range.

**Figure 7 ijms-22-01488-f007:**
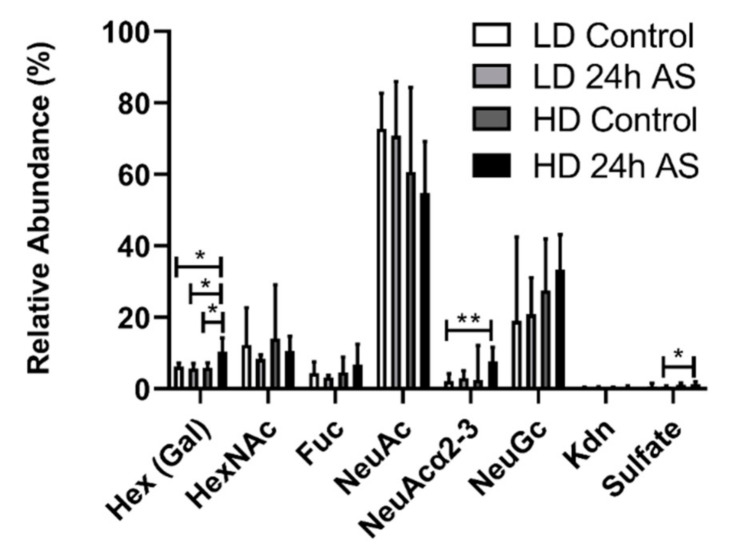
Relative abundance of structures with designated terminal moieties. Atlantic salmon were grown under low density (LD) and high density (HD) conditions with and without being subjected to an acute stress (AS) protocol and glycans analyzed by LC-MS. Statistics: * *p* < 0.05 and ** *p* < 0.01; Kruskal–Wallis test with Dunn’s post hoc test. Data are presented as median and interquartile range.

**Figure 8 ijms-22-01488-f008:**
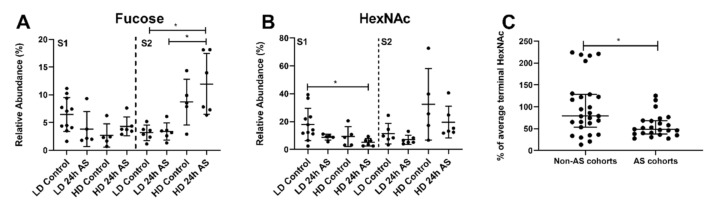
Stress-induced glycosylation changes that differed between sampling time points. Atlantic salmon were grown under low density (LD) and high density (HD) conditions with and without being subjected to an acute stress (AS) protocol and glycans analyzed by LC-MS. The first sampling time point (S1) was 73 days after the Atlantic salmon had been transferred to sea water and the second time point (S2) was 128 days after transfer. (**A**) Relative abundance of fucosylated structures and (**B**) Relative abundance of terminal HexNAc structures. (**C**) Effect of exposure to the acute stress protocol (AS) on HexNAc. To highlight the effect of the AS protocol regardless of fish density and time point, data were normalized to the median of the respective control groups (from panel (**B)**). Statistics: * *p* < 0.05, A and B: Kruskal–Wallis test with Dunn’s post hoc test and C: Mann–Whitney U-test. Data are presented as median and interquartile range.

**Figure 9 ijms-22-01488-f009:**
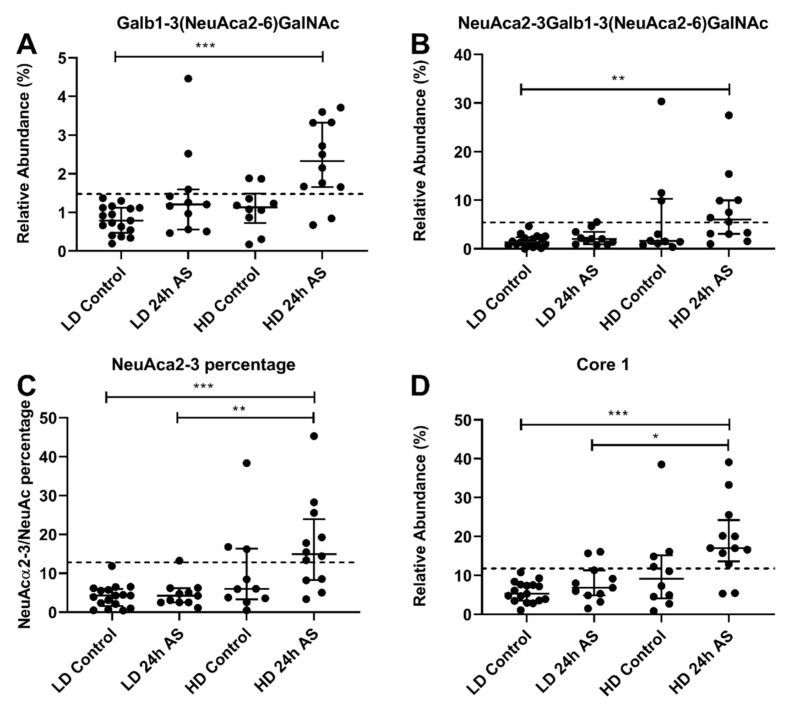
*O*-glycans with potential as stress biomarkers. Atlantic salmon were grown under low density (LD) and high density (HD) conditions with and without acute stress (AS) protocol and glycans analyzed my LC-MS. (**A**,**B**) Relative abundance of *m/z* 675b and 966 in mucins from individual fish from the different cohorts. (**C**) Percentage of NeuAcα2-3 of total relative abundance of NeuAc containing structures. (**D**) Relative abundance of structures composed of core 1. Statistics: * *p* < 0.05, ** *p* < 0.01, *** *p* < 0.001; Kruskal–Wallis test with Dunn’s post hoc test. Data are presented as median and interquartile range. The dotted lines indicate the highest value of the LD control group.

**Table 1 ijms-22-01488-t001:** Water quality parameters measured at the tank outlets on days 82, 116, and 172 of the experiment.

Day of Experiment	Treatment	Temp (°C)	O_2_ (% sat)	Conductivity (mS/cm)	Salinity (ppt)	pH	CO_2_ (mg/L)	TAN (mg/L)	Turbidity (NTU)
Day 82	LD	14.0 ± 0.0	91.7 ± 6.4	52.3 ± 0.1	33.8 ± 0.0	7.5 ± 0.1 *	12.8 ± 0.9 *	0.4 ± 0.1	0.2 ± 0.0
	HD	14.0 ± 0.1	93.0 ± 7.0	52.3 ± 0.0	33.8 ± 0.0	7.1 ± 0.2 *	22.1 ± 2.8 *	0.6 ± 0.1	0.3 ± 0.1
Day 116	LD	13.0 ± 0.0	90.0 ± 1.0	49.8 ± 0.0	31.9 ± 0.0	7.5 ± 0.2	10.7 ± 1.0 *	0.4 ± 0.1	0.2 ± 0.0
	HD	13.0 ± 0.1	92.7 ± 1.5	49.8 ± 0.0	31.9 ± 0.0	7.2 ± 0.2	16.8 ± 3.5 *	0.5 ± 0.1	0.2 ± 0.0
Day 172	LD	13.1 ± 0.0	94.0 ± 1.0	49.4 ± 0.0	31.7 ± 0.1	7.6 ± 0.2	12.3 ± 1.6	0.5 ± 0.0	0.2 ± 0.0
	HD	13.0 ± 0.1	97.0 ± 2.0	49.4 ± 0.0	31.7 ± 0.0	7.5 ± 0.1	14.0 ± 0.4	0.5 ± 0.0	0.2 ± 0.0

Values are given as tank means ± STDEV (*n* = 3 for each treatment) ^⁎^ Denotes significant difference (*t* test, *p* < 0.05) between treatments on days 82 and 116 of the experiment.

## Data Availability

MS/MS data with annotations on the tentative structures are available at https://unicarb-dr.glycosmos.org/references/396. Raw data files are available at https://glycopost.glycosmos.org/preview/755344085fef2fbaa7cf3, pincode: 6609.

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
