# Peer review of "Stress Impairs Skin Barrier Function and Induces α2-3 Linked N-Acetylneuraminic Acid and Core 1 O-Glycans on Skin Mucins in Atlantic Salmon, Salmo salar"

_ijms, 2021, doi:10.3390/ijms22031488_

Round 1

Reviewer 1 Report

This is an interesting and potentially valuable manuscript, but I suspect that a preliminary or rough draft was accidentally sent to me for review.   Figure 7 is absent and random font changes appear here and there (lines 11, 378, and 393).  Also, statistically insignificant differences (p>0.05) were discussed and interpreted as though they were biologically important and indicative or predictive of changes in health status (lines 20, 207, 249, 404).   I have seen papers rejected for the inclusion of biological arguments speculatively based on statistically insignificant “trends". 

Reviewer 2 Report

In this study the authors make a very tedious approach on how mucin glycosylation is affected by chronic and acute stress. Although sometimes AS treatment should be included so that differences between LD and HD make sense, some patterns and some quite useful results were obtained, following eventually to some strong candidate stress biomarkers. The experimental design, analysis of the results and their discussion were articulate and very meticulously presented by the authors. After addressing the following specific comments, I would be happy to endorse publication and I think that this work will be a very fine contribution to the current literature on such an interesting topic.

Line 70: In this paragraph, a brief overview about skin mucins is provided, but didn’t Padra et al. report that intestinal mucins promote A. salmonicida’s growth, whereas simple skin mucins do not?

Line 16: Maybe it is 50-80 for the high density as far as I understood from the Material and Methods section.

Lines 96-97: Maybe a small reference to the commercial densities would render the experimental design even more industry-focused.

Line 110: The subsets were from both low and high densities, right?

Line 133: It might be helpful to include the parameters that were examined for function assessment. Apart from TER, TEP, SCC and mannitol that are mentioned, are there any more?

Line 202: You could also state here the levels of significance that were used in the statistical analysis.

Line 206: low density without AS

Line 226: minor correction for *** or **** so that it matches legend and graph.

Line 242: blood pH B & D, blood HCO3- A & C. Also, please check again the legend and graph for *** and ****.

Line 204 and 246: I would recommend that the authors add a few explanatory information about the observed measurements in these two subsections, in a way similar to that of the following results subsections. For instance, they explain a bit of the background and biological meaning prior to or after the reported comparisons with comments such as those in lines 288-289, 306-307, 307-309 etc.

Line 332: I am afraid that Fig 7 is missing. Make sure you upload it in the revised version.

Line 387-392: the font of this part is different, please adapt it according to the body text.

Lines 393-394: I would suggest keep that in the results. If you insist on keeping it maybe it is better fit when you mention it in lines 497-498.

Lines 410: Are there any additional evidence in the literature to support or disprove the findings on mannitol, perhaps other sugars?

Line 458: This is very interesting and it seems that if the AS was omitted the results would look different. Therefore, am I right to deduce that the combination of chronic and acute stress is what mostly triggers the glycosylation responses. I think the authors more clearly in the later part of the discussion also state it too.

Lines 503-508: there is some level of repetition in this part and in lines 486-490. Consider merging them or maybe rephrasing and keeping one of them.

one figure is missing (Figure 7)

Round 2

Reviewer 1 Report

In my initial review I recommended strongly against publishing biological arguments based on comparisons that show no significant difference (p > 0.05).  It is inappropriate to refer to such comparisons as indicitave of "a tendency” (line 279) or to refer to such data as “a possible reduction in TER" (line 435).  TER was not reduced according to the authors' statistical analysis, which they do not seem inclined to fully accept.  p values above 0.05 do not indicate tendencies or possible reductions - they show no difference in response to a treatment.  

This is a good and valuable contribution with enough valid responses to demonstrate that stress was induced, in some cases with highly significant responses.  Changes detected in skin mucins are a very worthwhile observation with potential practical applicability.  It is a much weaker manuscript to attempt to stretch the outcome of the TER analysis to infer an effect when none is supported scientifically.  Stress responses are dynamic and it is possible or even likely to miss adaptive physiological parameters by sampling at one or two points. There may or may not be adaptive differences in TER but none have been shown with this experimental protocol.  It cheapens  the contribution to suggest that p = 0.07 suggests a difference.

The possibility of catecholaminergic mediation was not considered. 

Probability values are indicated with a lower case p throughout the manuscript, but in the current revision (line 435) P is shown inconsistenty in upper case.
